# Risk factors associated with cardiovascular disease among adult Nevadans

**Dieu-My T. Tran** [1]ᵒ*, **Nirmala Lekhak**[1]ᵒ, **Karen Gutierrez**[2], **Sheniz Moonie**[2]ᵒ

**1** School of Nursing, University of Nevada, Las Vegas, Las Vegas, Nevada, United States of America,
**2** School of Public Health, University of Nevada, Las Vegas, Las Vegas, Nevada, United States of America

ᵒ These authors contributed equally to this work.
* Dieu-my.tran@unlv.edu

## Abstract

### Objective

Cardiovascular disease (CVD) remains the number one cause of death in the US and Nevada is ranked 11th highest for CVD mortality. The study sought to examine the association between self-reported risk factors and CVD presence among adult Nevadans, between years 2011 and 2017.

### Methods

This is a cross-sectional, population-based study that utilized the 2011 and 2017 Nevada Behavioral Risk Factor Surveillance System data. Data were analyzed between 2019 and 2020.

### Results

A total of 5,493 and 3,764 subjects in 2011 and 2017, respectively were included. BMI (overweight/obesity) remained the most prevalent CVD risk factor. The second most common CVD risk factor was high cholesterol, followed by hypertension. Compared to females, males were 1.64 times more likely to have reported CVD in 2011, which increased to 1.92 in 2017. Compared to non-smokers, everyday smokers were 1.96 times more likely in 2011 and 3.62 times more likely in 2017. Individuals with high cholesterol status were 2.67 times more likely to have reported CVD compared to those with normal levels in 2011. In 2011, individuals with hypertension were 3.74 times more likely to have reported CVD compared to those who did not have hypertension. This relationship increased its magnitude of risk to 6.18 times more likely in 2017. In 2011, individuals with diabetes were 2.90 times more likely to have reported CVD compared to those without the condition.

### Conclusions

Public health and healthcare providers need to target preventable cardiovascular risk factors and develop recommendations and strategies locally, nationally, and globally.

**Funding:** The authors received no specific funding for this work.

**Competing interests:** The authors have declared that no competing interests exist.

## Introduction

Cardiovascular disease (CVD) remains the number one cause of death in the United States and globally. CVD accounts for 17.6 million deaths globally, which is an increase of 14.5% from 2006. During 2014 in the U.S., CVD and stroke accounted for an estimated 325.2 billion in direct and indirect costs, and this estimate is projected to increase [1]. Specifically, the prevalence of CVD in adults ages 20 years and older is 48% with increasing risk with advancing age in both males and females [1].

Multiple CVD risk factors such as smoking and high blood pressure place individuals at increased risk for CVD events. These risk factors tend to co-occur, thus further amplifying the risk [2]. Approximately half of American adults have at least one of three key cardiovascular risk factors (hypertension, hyperlipidemia, and smoking) despite the majority of these risk factors being preventable [3]. Hypertension is a significant risk factor for CVD, and it is the most prevalent risk factor. From 2011 to 2014, the prevalence of hypertension was 45.6% in adults using the new blood pressure classification (systolic blood pressure $\geq$ 130 mm Hg or diastolic blood pressure $\geq$ 80 mm Hg [4]. In contrast, it was 31.9% using the old guidelines (systolic blood pressure $\geq$ 140 mm Hg or diastolic blood pressure $\geq$ 80 mm Hg [5]. One of the strongest predictors of hypertension is the high intake of sodium [6]. According to a 2013, nationally representative sample of 827 adults, sodium intake was 4205 mg/d for males and 3039 mg/d for females, exceeding the recommendation of 2300 mg/d [1]. According to the American Heart Association (AHA) most recent guideline updates, among the cardiovascular risk factors, tobacco use remains a leading cause of preventable death in the United States and across the globe [1]. Tobacco use damages the heart and blood vessels, and nicotine raises blood pressure [3]. From 2013 to 2016, 26 million adults were diagnosed with diabetes mellitus, excluding the 9.4 million adults being undiagnosed. Adults who have diabetes have a higher risk of death from CVD compared to adults who do not have diabetes [3].

Additionally, a higher body mass index (BMI) was and continues to be associated with a significantly higher risk of CVD death. The prevalence of obesity was 39.6% among American adults [1]. However, literature supporting that a sedentary lifestyle is associated with an increased risk of cardiovascular events and mortality, less than 23% of adults report participating in inadequate amounts of physical activity meeting the 2008 federal guidelines [1]. One of the leading causes of the increase in sedentary activity is the increase in technology usage, especially smartphones.

Comparatively, Nevada is ranked 11th highest for CVD death rate in the U.S [1, 7]. In 2010, Nevadan adults who suffered from a heart attack were higher compared to the national average (5.0% versus 4.2%, respectively) [8]. Based on the 2017 data, 6,417 people died of CVD in Nevada [7]. When comparing Nevada CVD and stroke risk factors to the United States, Nevada reported a slightly higher prevalence of current smokers (17.6% versus 17.1%, respectively) as well as fewer individuals participating in more than 150 minutes of aerobic physical activity per week (65.7% versus 66.6%, respectively) [7]. However, Nevada has a lower prevalence of being overweight or obese (based on self-reported BMI) compared to the nation (65.7% versus 66.6%, respectively) [7]. It is of interest to explore the cardiovascular risk factors present among adult Nevadans and identify changes over the last decade to encourage proper funding and prevention/intervention of CVD. The purpose of this study was to examine the association between self-reported risk factors (heavy drinking, smoking, high cholesterol, hypertension, overweight/obesity, and diabetes) and CVD presence among adult Nevadans. A comparison of these factors between years 2011 and 2017 was assessed.

## Materials and methods

This is a cross-sectional, population-based study that utilizes the 2011 and 2017 Nevada Behavioral Risk Factor Surveillance System (BRFSS) data [9]. The BRFSS is an annual random digit dial (RDD) telephone survey that collaborates with the Centers for Disease Control and Prevention and each state across the Nation and U.S territories. Self-reported health conditions and risk behaviors are gathered across a representative sample of non-institutionalized adults 18 years and older for surveillance purposes. In an attempt to capture all segments of the population, landlines and cellular phones are incorporated. State health departments conduct telephone surveys, and they receive technical and methodological assistance from the CDC as needed. The landline and cell phone numbers selected for the study are obtained through random sampling. This survey consists of three components: the core component, optional modules, and state added content-specific questions. Nationwide and state-specific rates are available for comparison. Due to the selection and weighting procedures, rates are generalizable for each state and the nation. Data including sample weighting information are publicly available from the CDC website and available for download and analysis including the BRFSS Data User Guide [9]. Data analysis occurred from 2019 through 2020.

### Data analysis

Two dichotomous variables were combined to construct the binary dependent variable, self-report for cardiovascular disease (CVD) presence. This variable included the presence/absence of coronary heart disease (CHD) and/or myocardial infarction (MI). Participants were asked if a doctor, nurse, or other health professional had ever told them they have had coronary heart disease *or* myocardial infarction (MI). If the participant said yes to having *either* of these two conditions, they were considered to have CVD for analysis purposes. Independent variables were blocked in order to understand their respective contributions to the outcome of interest and included demographic variables (age, gender, race, and income), comorbid self-reported health conditions (cholesterol status, hypertension, diabetes, and BMI), as well as drinking, physical activity and smoking status. See Table 1 for the variables operational definitions.

Frequencies and weighted percentages for demographics were obtained overall, and the prevalence of select self-reported health variables by subgroup. Multiple logistic regression using the proc survey logistic procedure was utilized to model the association between the independent variables with CVD presence. Age-adjusted odds ratios and 95% confidence intervals were obtained. Age was a covariate in each model due to the strong association between age and CVD. Data were weighted in order to assure generalizability to the state of Nevada [9]. All analyses were conducted using SAS version 9.4.

## Results

### Demographics

The Nevada BRFSS dataset included a total of 5,493 and 3,764 subjects in 2011 and 2017, respectively. From both years, gender was evenly divided between females and males. Age was categorized into four groups: 18–24 years old, 25–44 years old, 45–64 years old, and 65 years and older. Over a third of the subjects in 2011 and 2017 were 25–44 years old (38.5% and 35.4%, respectively), white and non-Hispanic (58% and 53.5%, respectively) and with an income above $50,000 (37.5% and 44.8% respectively) (Table 2).

**Table 1. Operational definitions of variables.**

| Variables | Definition |
|---|---|
| **Age** | Categorized into four groups: 18–24 years old, 25–44 years old, 45–64 years old, and 65 years and older. For multiple logistic regression, age was categorized into 18–64 years old and 65 years and older. |
| **Gender** | Categorized into males versus females |
| **Race** | Categorized into five groups: White, non-Hispanic; Black, non-Hispanic; Hispanic; Other race, non-Hispanic; and Multiracial, non-Hispanic |
| **Income** | Categorized into five groups: <$15,000, $15,000-$25,000, $25,000-$35,000, $35,000-$50,000, and > $50,000 |
| **Cholesterol** | Adults who have had cholesterol checked & told by a doctor, nurse/other healthcare professional it was high |
| **Hypertension** | Adults who have been told they have high blood pressure, excluding pregnancy |
| **Diabetes** | Adults who have been told they have diabetes, excluding pregnancy |
| **BMI** | Calculated by the reported height and weight |
| **Heavy drinkers** | Men having >14 drinks per week and women having >7 drinks per week |
| **Physical activity** | Adults who met aerobic and strengthening guideline |
| **Smoking status** | Categorized into four groups: current smoker every day, current smoker some days, former smoker, and never smoked |
| **CHD or MI** | Have you ever been told you have CHD or MI, yes or no |

**Note**: BMI; body mass index; CHD; coronary heart disease; MI; myocardial infarction.

**Table 2. BRFSS descriptive statistics for Nevada by year.**

| | 2011 | | 2017 | |
|---|---|---|---|---|
| | n (N = 5,493) | Percent (%) | n (N = 3,764) | Percent (%) |
| **GENDER** | | | | |
| Male | 2241 | 50.6 | 1667 | 49.8 |
| Female | 3252 | 49.4 | 2097 | 50.2 |
| **AGE** | | | | |
| 18–24 | 290 | 11.7 | 228 | 11.3 |
| 25–44 | 1292 | 38.5 | 821 | 35.4 |
| 45–64 | 2238 | 33.6 | 1311 | 33.1 |
| 65+ | 1673 | 16.3 | 1404 | 20.2 |
| **RACE** | | | | |
| White, non-Hispanic | 4233 | 58.0 | 2668 | 53.5 |
| Black, non-Hispanic | 231 | 7.2 | 176 | 8.5 |
| Hispanic | 505 | 23.8 | 576 | 24.8 |
| Other race, non-Hispanic | 298 | 9.3 | 34 | 0.6 |
| Multiracial, non-Hispanic | 142 | 1.7 | 118 | 1.9 |
| **INCOME** | | | | |
| < $15,000 | 466 | 11.2 | 266 | 9.0 |
| $15,000–24,999 | 835 | 22.3 | 537 | 20.6 |
| $25,000–34,999 | 533 | 12.6 | 322 | 11.3 |
| $35,000–49,999 | 759 | 16.4 | 463 | 14.4 |
| $50,000+ | 2139 | 37.5 | 1523 | 44.8 |

## Gender and age vs. the cardiovascular risk factors outcome

Gender and age group differences in reported cardiovascular risk factors for both years were examined. Between males versus females, there were significant differences with the following: current smoker, hypertension, BMI, and CVD in 2011, which is similar in 2017 except cholesterol became significant between the genders (Tables 2 and 3). Specifically, between males and females, females tend to have higher risk factors reported in both years. Among the different age categories, there were significant differences between all the cardiovascular risk factors except for heavy drinkers in 2011 and 2017. Specifically, middle-aged (45–64 years old) reported more cardiovascular risk factors compared to the other age groups for both years. BMI (overweight and obese) remained the most prevalent CVD risk factor. The second most common CVD risk factor was high cholesterol (37.3% and 33.1%, respectively), followed by hypertension (30.8% and 32.6%, respectively). Among the CVD risk factors, heavy drinkers were the least common representing approximately 6% of the population. The cardiovascular risk factor prevalence was similar for both years when stratified by gender and age. Specifically, the prevalence was higher in males versus females between ages 18–64, except for diabetes. In 2011, more males reported diabetes, while in 2017, more females had diabetes. Additionally, diabetes was more prevalent in adults older than 65 and older than the remaining population. Lastly, in 2017, those 65 years and older exhibited an increase in all CVD risk factors relative to 2011 (Table 3).

## Relationships of modifiable and non-modifiable cardiovascular risk factors

Multiple logistic regressions were performed to examine the association of different demographic and risk factors with CVD presence in Nevada by year. Adjusted odds ratios are reported. Compared to females, males were 1.64 times more likely to have reported CVD in 2011 [OR = 1.64 (1.15, 2.33)], which increased to 1.92 in 2017 [OR = 1.92 (1.29, 2.85)]. As expected, the age group 18–64 years old, when compared with 65 years and older, were 83% less likely to have reported CVD in 2011 [OR = 0.17 (0.12, 0.25)], which dropped to 73% in

**Table 3. 2011 & 2017 Nevada BRFSS gender and age outcome breakdown.**

| 2011 | Males | Females | p-value | Age 18–24 | Age 25–44 | Age 45–64 | Age 65 + | p-value | Overall[*] |
|---|---|---|---|---|---|---|---|---|---|
| **Heavy Drinkers** | 170 (58.7%) | 214 (41.3%) | 0.055 | 23 (12.2%) | 84 (39.8%) | 171 (36.7%) | 106 (11.3%) | 0.107 | 384 (6.8%) |
| **Current Smoker** | 456 (56.4%) | 627 (43.6%) | 0.008 | 47 (8.9%) | 298 (45.2%) | 518 (36.4%) | 220 (9.5%) | <0.001 | 1,083 (22.9%) |
| **Cholesterol** | 796 (51.6%) | 1,100 (48.4%) | 0.264 | 9 — | 199 (24.1%) | 858 (44.9%) | 830 (29.5%) | <0.001 | 1,896 (37.3%) |
| **Hypertension** | 892 (55.4%) | 1,144 (44.6%) | 0.007 | 23 — | 187 (21.2%) | 856 (43.0%) | 970 (9.8%) | <0.001 | 2,036 (30.8%) |
| **BMI** | 1,536 (59.5%) | 1,623 (40.5%) | <0.001 | 105 (6.9%) | 713 (40.0%) | 1,390 (36.4%) | 951 (16.7%) | <0.001 | 3,159 (60.2%) |
| **CVD** | 247 (59.1%) | 246 (40.9%) | 0.029 | 0 — | 20 — | 170 (40.2%) | 303 (46.7%) | <0.001 | 493 (7.0%) |
| **Diabetes** | 267 (56.4%) | 295 (43.6%) | 0.135 | 3 — | 33 (15.5%) | 240 (47.0%) | 286 (25.3%) | <0.001 | 562 (10.3%) |
| 2017 | Males | Females | p-value | Age 18–24 | Age 25–44 | Age 45–64 | Age 65 + | p-value | Overall |
| **Heavy Drinkers** | 113 (57.1%) | 153 (42.9%) | 0.117 | 18 (10.8%) | 59 (44.2%) | 107 (31.5%) | 82 (13.5%) | 0.062 | 266 (6.2%) |
| **Current Smoker** | 271 (55.4%) | 295 (44.6%) | 0.070 | 16 (4.1%) | 137 (37.7%) | 253 (41.0%) | 160 (17.2%) | <0.001 | 566 (17.6%) |
| **Cholesterol** | 597 (53.9%) | 722 (46.1%) | 0.010 | 6 — | 127 (20.1%) | 522 (45.5%) | 664 (33.7%) | <0.001 | 1,319 (33.1%) |
| **Hypertension** | 692 (55.7%) | 779 (44.3%) | 0.001 | 11 — | 127 (16.2%) | 544 (44.9%) | 789 (37.5%) | <0.001 | 1,471 (32.6%) |
| **BMI** | 1,156 (57.8%) | 1,080 (42.2%) | <0.001 | 105 (8.1%) | 478 (35.6%) | 848 (35.8%) | 805 (20.5%) | 0.002 | 2,236 (65.7%) |
| **CVD** | 164 (61.0%) | 1,491 (49.0%) | 0.009 | 2 — | 13 — | 90 (40.6%) | 215 (49.8%) | <0.001 | 320 (6.8%) |
| **Diabetes** | 215 (48.7%) | 244 (51.3%) | 0.719 | 1 — | 25 (8.7%) | 183 (49.0%) | 250 (42.2%) | <0.001 | 459 (10.4%) |

Note.

[*]Overall non-grouped rate.—or N Excludes Unsure and No Response. BMI: Body Mass Index. CVD: Cardiovascular Disease.

2017 [OR = 0.27(0.18, 0.40)]. Among the different racial groups, Black non-Hispanics were 2.01 times more likely to have reported CVD in 2011 compared to White Non-Hispanic. Compared to Nevadans earning in the highest income level, those with an income level of less than $15,000 were 2.25 times [OR = 2.25 (1.34, 3.77)] and $15,000–25,000 were 2.06 times more likely [OR = 2.06 (1.19, 3.56)] to have reported CVD in 2011. Moreover, those in the income level of $25,000–35,000 were 2.30 times more likely to have reported CVD compared to those in the highest income level in 2017 [OR = 2.30 (1.19,4.42)] (S1 Table).

The study also examined the relationships of behavioral activities such as smoking, drinking alcohol, and physical activity. When adjusted for other risk factors, self-reported physical activity level was not significantly associated with CVD for either year. The relationship of heavy drinkers, when compared to non-heavy drinkers, changed from non-significant in 2011 to 77% less likely to have reported CVD [OR = 0.23 (0.08, 0.62)] in 2017. Compared to non-smokers, everyday smokers were 1.96 times more likely in 2011 [OR = 1.96 (1.16, 3.28)] and 3.62 times more likely in 2017 [OR = 3.62 (1.77, 7.44)] whereas former smokers were 2.12 times more likely in 2011 [OR = 2.12 (1.26, 3.57)] and 2.76 times more likely in 2017 [OR = 2.76 (1.76, 4.31)] to have reported CVD (Table 4).

The relationships between comorbid illnesses and CVD in Nevada were also examined. Individuals with high cholesterol status were 2.67 times more likely to have reported CVD compared to those with normal levels [OR = 2.67 (1.75, 4.07)] in 2011. However, this relationship was attenuated and lost significance in 2017. In 2011 individuals with hypertension were 3.74 times more likely to have reported CVD compared to those who did not have hypertension [OR = 3.74 (2.42, 5.77)]. This relationship increased its magnitude of risk to 6.18 times more likely in 2017 [OR = 6.18 (3.70, 10.33)]. The increase in the relationship of hypertension with CVD other factors in the model in 2017. In 2011 individuals with diabetes were 2.90 times more likely to have reported CHD or MI [OR = 2.90 (1.81, 4.65)] compared to those

**Table 4. Behavioral risk factors associated with CVD by year.**

| Calculated Variable | 2011 | | 2017 | |
|---|---|---|---|---|
| | OR | 95% CI | OR | 95% CI |
| **Heavy drinkers** [**] | | | | |
| No [*] | 1.00 | (1.00, 1.00) | 1.00 | (1.00, 1.00) |
| Yes | 0.68 | (0.37, 1.28) | **0.23** | **(0.08, 0.62)** |
| **Physical Activity** [¥] | | | | |
| Met both guidelines | 0.77 | (0.46, 1.29) | 0.63 | (0.39, 1.04) |
| Met aerobic guidelines only | 0.57 | (0.37, 1.29) | 0.64 | (0.40, 1.03) |
| Met strengthening guidelines only | 0.83 | (0.31, 0.88) | 0.64 | (0.29, 1.41) |
| Did not meet either guideline [*] | 1.00 | (1.00, 1.00) | 1.00 | (1.00, 1.00) |
| **Smoker Status** | | | | |
| Current smoker–every day | **1.96** | **(1.16, 3.28)** | **3.62** | **(1.77, 7.44)** |
| Current smoker–some days | 1.41 | (0.67, 3.00) | 2.17 | (0.96, 4.90) |
| Former smoker | **2.12** | **(1.26, 3.57)** | **2.76** | **(1.76, 4.31)** |
| Never smoked [*] | 1.00 | (1.00, 1.00) | 1.00 | (1.00, 1.00) |
| **Age** | | | | |
| 18–64 | **0.20** | **(0.13, 0.31)** | **0.24** | **(0.16, 0.35)** |
| 65 or older [*] | 1.00 | (1.00, 1.00) | 1.00 | (1.00, 1.00) |

[*]Reference group.

[**]Men having >14 drinks per week and women having >7 drinks per week.

¥: Adults who met aerobic and strengthening guidelines.

without the condition. The strength of this relationship decreased to 1.89 more likely in 2017 [OR = 1.89 (1.18, 3.05)]. Compared to average weight, those of elevated BMI levels did not demonstrate a significant increase of CVD; however, underweight individuals were 3.30 times more likely to have reported CVD [OR = 3.30 (1.29, 8.44)] in 2011. While in 2017 underweight individuals were 89% less likely to have reported CVD compared to those of average weight [OR = 0.11 (0.02, 0.55)]. Notably, hypertension was the leading cardiovascular risk factor when compared to cholesterol, diabetes, BMI, and age (Table 5).

## Discussion

The main aim of the study was to find the relationship between different cardiovascular risk factors and CVD among adult Nevadans in 2011 and 2017. Thus, the study provides an over-view of how cardiovascular risk factors among this population has changed in 6 years. The study found some anticipated results as well as other interesting relationships and trends. As expected, males had higher odds of reporting CVD than females, and this increased between years. Interestingly, the literature also demonstrated that adult males had a higher risk of CVD compared to females; however, the gender gap closes as females gets older [10, 11]. Similarly, the younger age group had less likelihood of reporting CVD in 2011, compared to the older age group. Although the younger age group still had less likelihood of reporting CVD in 2017, this percentage was less compared to 2011 indicating increased prevalence among the 18–64 years age group. Andersson and Vasan's review indicated that CVD rates are increasing among younger adults, which could be due to high prevalence of obesity and lack of exercise and nutritious food [12]. As this population grow older, the healthcare industry needs to be better prepared and educated to care for them. Among racial groups, as expected, Black Non-Hispanics were more likely to have reported CVD compared to White Non-Hispanics; how-ever, there were no significant racial differences in reported CVD in 2017. Because data

**Table 5. Comorbid risk factors associated with CVD by year.**

| Calculated Variable | 2011 | | 2017 | |
|---|---|---|---|---|
| | OR | 95% CI | OR | 95% CI |
| **Cholesterol Status [a]** | | | | |
| No, checked but not been told high * | 1.00 | (1.00, 1.00) | 1.00 | (1.00, 1.00) |
| Yes, checked and been told high | **2.67** | **(1.75, 4.07)** | 1.46 | (0.94, 2.27) |
| **Hypertension Status [b]** | | | | |
| No * | 1.00 | (1.00, 1.00) | 1.00 | (1.00, 1.00) |
| Yes | **3.74** | **(2.42, 5.77)** | **6.18** | **(3.70, 10.33)** |
| **Diabetes Status [c]** | | | | |
| No * | 1.00 | (1.00, 1.00) | 1.00 | (1.00, 1.00) |
| Yes | **2.90** | **(1.81, 4.65)** | **1.89** | **(1.18, 3.05)** |
| **Age** | | | | |
| 18–64 | **0.39** | **(0.26, 0.58)** | **0.43** | **(0.28, 0.67)** |
| 65 or older * | 1.00 | (1.00, 1.00) | 1.00 | (1.00, 1.00) |
| **Adult BMI** | | | | |
| Underweight | **3.30** | **(1.29–8.44)** | **0.11** | **(0.02, 0.55)** |
| Normal weight * | 1.00 | (1.00, 1.00) | 1.00 | (1.00, 1.00) |
| Overweight | 0.90 | (0.59, 1.36) | 1.36 | (0.85, 2.18) |
| Obese | 0.75 | (0.42, 1.34) | 1.23 | (0.70, 2.17) |

*Reference group.

collected were cross-sectional in nature, the population was different at these two-time points. However, the random digit dialing methodology and weighting protocol for this survey assure that the findings are representative of the state for any given year. With more focus on reducing health disparity, this finding is a positive and expected outcome. However, it will be interesting to see if this trend continues in the future and if any preventive measures exist to reduce the burden of CVD in Nevada. There has been documented evidence in the literature supporting the use of preventive measures to reduce CVD, which the AHA reports summarized [1, 13]. In 2011, lower income was associated with a higher odds of reporting CVD compared to a higher income level. However, in 2017, Nevadans reporting less than $15,000 annual income continued to have higher likelihood of reporting CVD with a slight increase in odds relative to 2011 and income group of $25–35,000 became significant in terms of increased risk compared to $50,000 or higher income level group. The increased odds in the lowest income level attenuated and were no longer significant in 2017. Overall, Nevadans with lower earnings had higher odds of self-reported CVD. This is consistent with the literature as lower socioeconomic status is related to lack of access to quality health care, which in turn increases the likelihood of elevated CVD prevalence and higher hospital readmission rates [14, 15].

Along with the relationship of different demographic factors, the study also looked at modifiable cardiovascular risk factors (behavioral and health conditions) in Nevada. In order to reduce bias, age was adjusted for in each model due to the strong and linear relationship between age and having a heart event. Among different behavioral factors, it is interesting to note that physical activity was not significantly associated with CVD in both years. This could be explained by the limitations of self-report data instead of objective measurements. This is in contrast to current literature, which indicates that any kind of physical activity is beneficial to cardiovascular health [16]. The non-significant relationship between heavy drinking and CVD was significant in 2017. In 2017, heavy drinkers were less likely to have reported CVD when compared to non-heavy drinkers. In this study, alcohol consumption demonstrated a slight protective effect on cardiovascular health. Research has indicated that alcohol shows a U-shaped relationship with CVD, indicating that low or moderate drinking may be beneficial for cardiovascular health concerning abstinence or abusive drinking [17]. Protective characteristics of moderate alcohol drinking are associated with its blood-thinning properties, reduced inflammatory response and increased high-density lipoprotein [18]. Compared to non-smokers, regular smokers, and former smokers were at greater risk. This risk was higher for everyday smokers and the odds increased in 2017 for both of these groups. These data did not provide further explanation on who the former smokers were and when they had the heart event; therefore, temporality could not be established. Former smokers may have experienced the heart event prior to quitting. Smoking continues to remain one of the modifiable risk factors of CVD although there has been significant improvement in preventive measures to reduce CVD and smoking cessation programs [19]. Although smoking use in Nevada continues to decline, there is still initiatives to prevent secondhand exposure as evident by continuous funding from CDC to prevent and control smoking [20].

Among different health conditions that were examined, hypertension had the most substantial relationship to self-reported CVD. This relationship increased from 3-fold in 2011 to over 6-fold risk in 2017, causing the other comorbidities to attenuate or become insignificant. Nevadans with diabetes were at greater risk for both years. Nevadans with elevated cholesterol status had higher odds of reporting CVD in 2011, yet this relationship was not significant in 2017. This could be due to the dominating relationship of hypertension with the outcome. This is in support of the literature as hypertension is one of the significant documented cardiovascular risk factors in the literature, along with obesity [21, 22]. It is important to note that hypertension rates will increase partially due to the new blood pressure classification

guidelines implemented for tighter blood pressure control [4]. There has been lot of research indicating that obesity is one of the modifiable risk factors of CVD [1]. However, in our study obesity was not a significant predictor of CVD. There have been studies to support that the risk of CVD associated with obesity is mainly due to comorbid conditions [23, 24]. This is reflected in our findings where comorbid conditions such hypertension and diabetes are better predictors of CVD. One of the interesting findings of this study is that compared to normal weight, individuals who were underweight had reported higher odds of CVD in 2011 and lower risk in 2017. Further analyses are needed to understand this interesting association. However, fewer research in underweight and CVD has shown contrasting findings. Research indicating increased CVD risk with underweight have associated this risk with lack of proper nutritional status and those with lower CVD risk have associated underweight with lower cholesterol and without comorbid conditions such as diabetes and hypertension [25, 26].

## Limitations

This study is not without limitations. First, this study's assessment and variables relied on self-report; therefore, misclassification is a possibility due to recall bias. We also lack objective measurements in physical activity, as well as clinical markers such as fasting lipid panels, glucose, and blood pressure readings taken by a trained professional. Another limitation is the lifetime prevalence of CVD, it is quite possible that participants have made lifestyle changes post-heart events. This would explain some unexpected findings in the study, such as the insignificant association between physical activity and cardiovascular condition. Thirdly, despite this study being a population-based research design, it is cross-sectional; therefore, the sample changed over time. However, using the BRFSS dataset provided a large sample size of over 9000 participants with a better representation of the state. Generalizability can be made because of the BRFSS weighting and inclusion of random individuals that eliminate selection biases.

## Implications and conclusions

To reduce CVD, public health and healthcare providers need to target preventable cardiovascular risk factors and develop recommendations and strategies. Currently, locally and nationally, strategies and efforts are in place to reduce the CVD burden. This includes the AHA recommendations of the *Life's Simple 7 (*stop smoking, eat better, get active, lose weight, manage blood pressure, control cholesterol, and reduce blood sugar) and Healthy People 2020 goals and objectives to increase overall cardiovascular health in the United States population which have achieved little success. Specifically, this study identified hypertension as the leading cardiovascular risk factor, which warrants a closer examination of our current practice in clinical, research, and policy to reduce prevalence statewide and nationally. The long-term benefit of addressing modifiable cardiovascular risk factors such as hypertension before heart health declines may be impactful, including improved quality of life as well as decreases in the economic burden to the health care system. When it comes to preventable risk factors, primary prevention should be a priority locally, nationally, and globally.

## Supporting information

**S1 Table. Sociodemographic factors associated with CVD by year.**
(DOCX)

## Author Contributions

**Conceptualization:** Dieu-My T. Tran, Nirmala Lekhak, Sheniz Moonie.

**Formal analysis:** Karen Gutierrez, Sheniz Moonie.

**Funding acquisition:** Dieu-My T. Tran.

**Methodology:** Sheniz Moonie.

**Validation:** Dieu-My T. Tran.

**Writing – original draft:** Dieu-My T. Tran, Nirmala Lekhak, Sheniz Moonie.

**Writing – review & editing:** Dieu-My T. Tran, Nirmala Lekhak, Sheniz Moonie.

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
