## [Decision Letter · Decision Letter 0]

22 Dec 2020

PONE-D-20-26195

Behavioral Risk Factors Associated with Cardiovascular Disease among Adult Nevadans

PLOS ONE

Dear Dr. Tran,

Thank you for submitting your manuscript to PLOS ONE. After careful consideration, we feel that it has merit but does not fully meet PLOS ONE’s publication criteria as it currently stands. Therefore, we invite you to submit a revised version of the manuscript that addresses the points raised during the review process.

We look forward to receiving your revised manuscript.

Kind regards,

Sabine Rohrmann

Academic Editor

PLOS ONE

Journal Requirements:

'NO - The funders had no role in study design, data collection and analysis, decision to publish, or preparation of the manuscript.

Acknowledgements: The publication fees for this article were supported by the UNLV University Libraries Open Article Fund (if accepted for publication).'

Reviewers' comments:

Reviewer's Responses to Questions

**Comments to the Author**

1. Is the manuscript technically sound, and do the data support the conclusions?

Reviewer #1: Yes

Reviewer #2: Yes

2. Has the statistical analysis been performed appropriately and rigorously? 

Reviewer #1: No

Reviewer #2: Yes

3. Have the authors made all data underlying the findings in their manuscript fully available?

Reviewer #1: Yes

Reviewer #2: Yes

4. Is the manuscript presented in an intelligible fashion and written in standard English?

Reviewer #1: Yes

Reviewer #2: Yes

5. Review Comments to the Author

Reviewer #1: The purpose of this cross-sectional study is to examine the association between self-reported risk factors (heavy drinking, smoking, high cholesterol, hypertension, overweight/obesity, and diabetes) and CVD presence among adult Nevadans. However, some issues need to be clarified or revised:

1) Some of the self-reported risk factors (cholesterol, hypertension, diabetes) are not behavioral risk factors. The title of this manuscript is improper.

2) Several variables such as hypertension, diabetes, high cholesterol, obesity (BMI) and smoking are not clearly defined. Please give details.

3) The description of methodology and data analysis is not rigorous enough. For example, the raking weighting methodology should be more specific including the weighted variables. The BRFSS Data User Guide of CDC should also be referenced.

4) Please show the data in line122-134 in a table format.

5) 95% confidence intervals (CIs) and the missing number of each subgroup are necessary in Table1, 2 and 3.

6) In order to avoid ambiguity, weighted percentages should also be illustrated below each table.

7) Some data in Table 2 and 3 do not have percentages. (e.g., 9--,23--,20--, etc) Please illustrate the meaning below each table. Meanwhile, please add the abbreviations.

8) There are several data errors in these tables. Please recheck them carefully. (e.g., the percentage of female current smoker in Table 3 is15.8%; number of male CVD in Table 3 is 164, etc)

9) Nineteen references may be insufficient. Further discussion is needed instead of retelling the results.

10) Please do not use semicolons in sentences！

11) Several minor errors should be revised as follow: “mg/dL” should be “mg/d” in line 41; some brackets are italic; “0.23 0.08, 0.62)” should be “0.23 (0.08, 0.62)” in line 139.

Reviewer #2: 1. The study examined the association between risk factors and CVD presence among adult Nevadans between 2011 and 2017, but it is less objective since the study relied on the self-reporting.

2. Multiple logistic regressions were performed to examine the association of different demographic and risk factors with CVD presence in Nevada by year. Please add a table showing all the result, so difference would be obvious between 2011 and 2017.

3.Age was categorized into four groups: 18-24 years old, 25-44 years old, 45-64 years old, and 65 years and

100 older. Could you please give a reason of age categories in this study.

4.BMI (overweight and obese) remained the most prevalent CVD risk factor. However, compared to normal weight, there was huge difference of underweight individuals on reporting CVD(Table 5). Please provide some further discussion on it.

6. PLOS authors have the option to publish the peer review history of their article (what does this mean?). If published, this will include your full peer review and any attached files.

Reviewer #1: No

Reviewer #2: No

---

## [Author Response · Author response to Decision Letter 0]

28 Jan 2021

Please refer to our Response to Reviewer Table that addressed all the reviewers' comments/feedback.

---

## [Editor Report · Decision Letter 1]

2 Feb 2021

Risk Factors Associated with Cardiovascular Disease among Adult Nevadans

PONE-D-20-26195R1

Dear Dr. Tran,

We’re pleased to inform you that your manuscript has been judged scientifically suitable for publication and will be formally accepted for publication once it meets all outstanding technical requirements.

Kind regards,

Sabine Rohrmann

Academic Editor

PLOS ONE
---

## [Editor Report · Acceptance letter]

4 Feb 2021

PONE-D-20-26195R1 

Risk Factors Associated with Cardiovascular Disease among Adult Nevadans 

Dear Dr. Tran:

I'm pleased to inform you that your manuscript has been deemed suitable for publication in PLOS ONE. Congratulations! Your manuscript is now with our production department. 

Kind regards, 

on behalf of

Dr. Sabine Rohrmann 

Academic Editor

PLOS ONE